

# The Mini Nutritional Assessment tool's applicability for the elderly in Ethiopia: validation study

Megersso Urgessa

Department of Public Health, School of Health Sciences, Madda Walabu University, Shashemene, Oromia, Ethiopia

## ABSTRACT

**Background**. The Mini Nutrition Assessment (MNA) is a widely used and valid tool for screening and assessment of malnutrition among the elderly population worldwide. However, MNA has not been validated among the Ethiopian elderly population and this study assessed the validity of the tool for the target population.

**Methods**. Cross-sectional validation study design employed to validate MNA in Meki town, East Ethiopia. This study included 176 randomly selected elders living in the community, whereas amputated, bedridden, visible deformity, known liver and/or renal disorders were excluded. The original MNA questionnaires were translated to local language and administered to each participant after doing the pretest. The anthropometric, self-perception of nutritional status and serum albumin concentrations were measured. Reliability, validity, sensitivity, specificity, Positive Predictive Value (PPV), and Negative Predictive Value (NPV) were calculated. Receiver-operating characteristic (ROC) curve analysis was plotted to identify the area under the curve (AUC) and optimal cut-off value for the prediction of malnutrition.

**Result**. A total of one hundred and seventy-six elders participated in this study. Of the total participants, 78(44.3%) were males. The mean (SD) age of the participants was 67.6 ($\pm$5.8) years and ranged from 60 to 84 years. The prevalence of malnutrition based on the MNA criteria (MNA < 17 points) was 18.2%, and 13.1% based on serum albumin concentration (<3 g/dl).The MNA had an overall Internal consistency of Cronbach's alpha 0.61. The tool also demonstrated significant criterion-related validity (0.75, $p < 0.001$) and concurrent validity (0.51, $p < 0.001$) with serum albumin concentration and self-perception of nutritional status respectively. Using the original cut-off point, the sensitivity, specificity, PPV and NPV of the tool were 93.5%, 44.6%, 65.4% and 86.0%, respectively. By modifying, the cut-off point to a value of <20.5, the sensitivity and specificity of the tool increases to 97.6% and 82.8% respectively. The AUC (95%CI) showed an overall accuracy of 92.7% (88.5, 96.9).

**Conclusion** . The MNA tool can be used as a valid malnutrition screening tool for the Ethiopian elderly population by modifying the original cut-off point.

Corresponding author
Megersso Urgessa,
megurgessa@gmail.com

## BACKGROUND

Elderly people refer to those who are 60 years and above (*Ethiopia Ministry of Labor and Social Affairs, 2013*; *United Nations, 2019*), and currently it is increasing at a faster rate. Every second two persons celebrate their 60th birthday globally. By 2050 the elderly population is expected to double in the world (*United Nation Population Fund, 2012*). In Europe alone, the elderly population will constitute about thirty-four percent of the entire population by 2050 (*Chatterji et al., 2015*). Even in developing countries like Ethiopia elderly populations are rising, and they represent about 3.3% (3.3 million) of the 110 million population, with 4.42% of the total population living in the Urban area (*Ethiopia Ministry of Labor and Social Affairs, 2013*). In addition, the country's life expectancy has increased to 67.8 years (*Ethiopia Population Census Commission, 2014*; *Government of Ethiopia, 2022*). Obviously, with aging the elderly population's risk of developing communicable and non-communicable diseases increases (*Hayflick, 2007*). Hence, maintenance of optimum nutrient consumption in these age groups is of paramount importance to prevent diseases (*Russell et al., 2013*). Especially in this century, elderlies are prone to the dual burden of malnutrition; under- nutrition or over-nutrition (*WHO, 2021*), and chronic non-communicable diseases (*Blossner, De Onis & Prüss-Üstün, 2005*; *Brownie, 2006*; *HelpAge Intrnational, 2013*).

Protein-energy malnutrition, a condition resulting from inadequate consumption of nutrients (*Cederholm et al., 2015*), is a specific concern in the elderly population because it is associated with increased morbidity and mortality (*Skates & Anthony, 2012*). The magnitude of malnutrition varies from setting to setting. In developed countries prevalence of malnutrition is reported to be 15%, among community members, 23–62% in hospital settings, and more than 80% in intensive care units (*Morley, 1997*). In developing countries like South Africa, for instance, the prevalence of malnutrition is reported to be 50% in hospital settings (*Charlton, Kolbe-Alexander & Nel, 2007*). The figure is more or less similar in Chile, where the prevalence is 58% among the hospital population (*Urteaga, Ramos & Atalah, 2001*).

In Africa, among community populations, the prevalence is reported to be 26.5% in Egypt (*Hamza et al., 2018*), and 28.3% in Ethiopia (*Hailemariam, Singh & Fekadu, 2016*). Given the elderly population's increasing population size and risk of malnutrition; it is crucial to devise methods of early detection. For effective screening and detection of malnutrition, a valid and reliable malnutrition screening tool is necessary (*Eglseer, Halfens & Lohrmann, 2017*). This further assists those elders who need intervention (*Skipper et al., 2012*). Malnutrition screening tools are mostly easy to administer and contain structured questionnaires that include questions related to the difficulty of chewing, appetite loss, or functional limitations. The tools also enable documentation of indicators of malnutrition, like involuntary weight losses (*Kondrup et al., 2003*). However, the validity of these tools is very crucial to carry out the screening process so that one can measure what it is intended to measure as far as malnutrition is concerned (*Skates & Anthony, 2012*; *Jones, 2004*).

There are different valid screening tools used to screen malnutrition among geriatrics, and the Mini nutrition assessment (MNA) is the most widely used (*Secher et al., 2007*).
This tool was developed in the early 1990s and published in 1994 (*Guigoz, 1994*). It is a short and simple tool that takes 10–15 min to complete (*Nestlé Nutrition Institute, 2022b*). It has 18-items with four categories (anthropometric assessment, dietary assessment, global assessment, and subjective assessment). All the eighteen items attribute to a score with a maximum of 30-points. Based on the final score it categorizes the population into three groups: malnutrition if the score is <17 points, at risk of malnutrition, for scores between 17–23.5 points, and well-nourished, if the score is between 24 and 30 points, inclusive (*Nestlé Nutrition Institute, 2022a*).

It is the only nutritional screening and assessment tool that incorporates functionality, mobility, and depression (*Anthony, 2008*; *van Bokhorst-de van der Schueren et al., 2014*). Moreover, it is reliable, inexpensive, does not require laboratory investigation, and is used in all settings (*Guigoz, 1994*; *Guigoz, 2006*). It is also able to detect risks of malnutrition before the severe change in individuals' weight or serum albumin occurs (*Guigoz, 2006*). It also correlates with serum albumin concentration (*Vellas et al., 2000*). Reports also indicated that it predicts mortality and length of stay in hospital (*Kagansky, 2005*). There are hundreds of proteins circulating in plasma and serum albumin is one. To measure this one needs a serum fluid that remains after plasma has clotted, fibrinogen, and most of the clotting factors removed (*Busher, 1990*; *John, Hall & Guyton, 2011*). The normal range of protein is 6.5−8.5 g/dl (*Tracey, 2005*; *WHO, 2000*) and out of this albumin accounts large proportion (50–60%), with a normal value ranging from 3.5–5 g/dl (*Tracey, 2005*; *WHO, 2000*). It has a half-life of 20 up to 22 days. Whereas its precursor pre albumin (transthyretin) has only 2 to 4 days (*Smith, 2017*). A systematic review of literature conducted by Zhang and colleagues in 2017, recommended the use of albumins and other biomarkers including pre- albumin, hemoglobin, total cholesterol and total protein for the elderly's nutritional assessment, regardless of body's inflammation status (*Zhang et al., 2017*). The pre-albumin (transthyretin), retinol-binding protein and transferring are markers of short-term nutritional status (*Victor et al., 2009*). Serum albumin is also used as a predictor of morbidity and mortality in elderly people (*Simon, 2009*). Based on serum level of albumin nutritional status of elderly population can be categorized as malnutrition if <3.0 g/dl, at risk if 3 to 3.5 g/dl, and well-nourished if >3.5 to 5 g/dl (*Rodrigueza et al., 2018*; *Bharadwaj et al., 2016*).

Even though MNA is validated and used in a different country, it is not readily applicable to other countries. In part this is due to varying characteristics of the population's anthropometric measurement and nutritional characteristics; from one setting to the other. For instance, MNA was not applicable in the Chilean population (*Urteaga, Ramos & Atalah, 2001*). The original cut-off value was also not reliable for Irian elders (*Amirkalali et al., 2010*), and Japan's population as well (*Kuzuya et al., 2005*). In Ethiopia, MNA has not been tested on the elderly population and there is a gap of established cut-off points, to screen and assess malnutrition. Therefore, this study attempted to validate MNA using serum albumin concentration as a golden standard in the Ethiopian geriatric population.

## METHODS

### Participants

The study was conducted in Meki town, Eastern part of Ethiopia from March to April 2020. Initially, we conducted a house-to-house survey to estimate the total number of elderly people (aged 60 and above) living in the setting. Each were given a unique identifier to help us develop a sampling frame. At this stage, we have also secured contact information to make data collection smooth. Following this, we calculated the sample size needed using BUNDER'S FORMULA (*Buderer, 1996*), and our calculation yielded 176 study participants. Recruitment was then followed afterward using a computer-generated simple random sampling technique. Using the unique identifier and the contact information we have secured at the earlier stage, from our sampling frame we have approached those elders otherwise healthy, do not have any signs of deformity, amputation, not incapacitated, do not have known liver and kidney disorders. We have then presented detailed information about the nature of the study, and after consent was provided, detailed data were obtained from the individual.

### Nutritional assessment

A human blood sample (4 mL) was collected in the morning before 9:30 am, after a full overnight fast, using a cupper-and zinc-free syringe. Serum albumin concentration was measured by automated Bromocresol green method using BCG reagent and its standard manufactured by Jourilabs (https://www.jourilabs.com/). All samples were handled according to WHO guidelines on standard operating procedures for clinical chemistry (*WHO, 2000*), and reagent with its standard manufacturer order (https://www.jourilabs.com/). It classifies as malnutrition if score is <3.0 gram/deciliter (g/dl), at risk of malnutrition if score is 3 to 3.5 g/dl, and well-nourished for score between 3.5 to 5 g/dl (*Vellas et al., 2000*; *Rodrigueza et al., 2018*; *Bharadwaj et al., 2016*).

Pre-tested Original MNA questionnaires [see Additional file 1] were administered to all participants. The MNA® was used in accordance with Nestlé's terms and conditions (*Nestlé Nutrition Institute, 2022a*). All participants' weight, height, Mid-upper arm circumference (MUAC), and calf-circumference (CC) were measured twice, and the average record was used for this study. Height was measured using a stadiometer (Seca 213, Germany), participant bare feet, with their buttock, heels, and occiput touching the board. Participants' height was recorded to the nearest 0.1centimeters (cm). Weight was recorded to the nearest 0.1 kg; using calibrated digital scales placed on a hard flat surface with subjects in light clothes and bare feet. The weighing scale was checked after each measurement with a 2 kg standard weight. MUAC was recorded to the nearest 0.1 cm and was measured at the mid-point, between the tip of the Acromion and Olecranon process on the back of the upper arm while the subject's forearm held a freely horizontal position. CC was measured at the widest circumference between ankle and knee and was recorded to the nearest 0.1 cm, using a flexible tape in a sitting position, with a leg 90-degree (90°) at the knee. Body mass index (BMI) was computed as body weight in kilograms divided by the squares of height in meters. All data were collected by trained Nurses and laboratory professionals.

## Data processing and analysis

The data were first entered into Epidata version 3.1, then exported to and analyzed by the IBM Corp. Released 2017. IBM SPSS Statistics for Windows, Version 25.0 Armonk, NY: IBM Corp.

Variables of interest were described using means, standard deviations (SD), frequencies, and percentage (%). A $P$-value <0.05 was used to define statistical significance. To assess the reliability of the MNA instrument in the population, the overall internal consistency was evaluated by Cronbach's alpha. Usually, an alpha value of $\geq$ 80 is considered good, however, a value of 0.70–80 and 0.60–0.70 is adequate and acceptable respectively (*Garson, 2016*). Also, Spearman's rank correlation coefficient for all the 18-items was calculated as well.

To calculate the criterion-related validity of the tool by comparing it with the gold standard (*Streiner, Norman & Cairney, 2015*), we have assessed the correlation (Spearman's rho) coefficient between MNA and serum albumin concentration. A value of 0.90–1.00 is considered very high and $\leq$ 0.50 low, whereas, values of 0.50−0.70 and 0.70–0.90 are considered moderate and high respectively (*Mukaka, 2012*). Concurrent validity (*Streiner, Norman & Cairney, 2015*), was assessed again using Spearman's rank coefficient between MNA items and participants' self-perception of nutritional status. In addition, we assess Concurrent validity using the correlation between MNA and BMI classification of nutritional status.

Agreement between the two methods the MNA tool and serum albumin concentration was assessed by calculating a weighted kappa coefficient. A kappa value of 0.80−1.0 is considered perfect agreement (*Landis & Koch, 1977*). The sensitivity, specificity, positive predictive value (PPV), and negative predictive value (NPV) were calculated using serum albumin concentration as a golden standard.

The markers of malnutrition considered for the analysis are if the MNA score is <24 points or serum albumin concentration is <3.5 g/dl. A receiver operating characteristic curve (ROC) was plotted using serum concentration <3.5 g/dl as a marker of malnutrition. The area under the ROC curve (AUC) was evaluated to determine the overall accuracy of the MNA tool and usually, a value $\geq$ 0.9 is considered excellent (*Zeng & Wang, 2010*). The optimal cutoff value was calculated using Youden's J index (sensitivity + specificity -1) (*Youden, 1950*).

## Ethical review and participants consent

This study was conducted according to the guidelines laid down in the World Medical Association (WMA) Declaration of Helsinki and all procedures involving research study participants were reviewed and approved by Jimma University, Institute of Health, Ethical review committee (ERC) with ref no IRB 0063/2020. Written informed consent was obtained from all participants.

## RESULTS

A total of 176 elders participated in the study, and the response rate was 100%. The mean (SD) age of the participants was 67.86 ($\pm$5.8) years and 98 (55.7%) were females. Overall,

**Table 1** Characteristics of study participants elderly people aged 60 and above years in the community, Meki town, East Ethiopia, 2020.

| Variable | n (%) |
|---|---|
| Gender | |
| Male | 78 (44.3%) |
| Female | 98 (55.7%) |
| Age category in year | |
| 60–64 | 61 (34.7%) |
| 65–69 | 63 (35.8%) |
| 70–74 | 24 (13.6%) |
| 75–79 | 23 (13.1%) |
| ≥ 80 | 5 (2.8%) |
| | **mean, SD** |
| Age in year | 67.6 (5.79) |
| Weight in Kg | 70.7 (10.15) |
| Height in meters | 1.7 (0.07) |
| Serum albumin score in g/dl | 3.7 (0.60) |
| MNA (sum score) | 20.7 (3.46) |

mean (SD) of the total MNA score was 20.7(3.5), and the mean (SD) of the serum albumin concentration was 3.7 (0.60) (Table 1).

The internal consistency of the MNA tool was adequate (Cronbach $\alpha = 0.61$). Homogeneity between the eighteen MNA items was adequate with Cronbach's Alpha of 0.61. Cronbach's alpha if an item deleted ranged from 0.526 to 0.633. The Scale's Cronbach's alpha would be 0.633 if the acute stress item were removed from the scale (Table 2). But, in this study, no item was removed and we used Cronbach $\alpha = 0.61$ throughout the analysis. In addition, MNA's total score significantly correlates with all its items ($r_s > 0.242$, $P < 0.05$).

Criterion-related validity of the MNA tool was tested by correlating it with the serum albumin concentration, and the result was statistically significant ($r_s = 0.746$, $P < 0.05$). Similarly, the concurrent validity of the tool was calculated by correlating the total scores of the item with the BMI classification of nutritional status and the result was again significant ($r_s = 0.392$, $P < 0.05$). Similarly, the MNA tool and self-perceived nutritional status correlated significantly ($r_s = 0.514$, $P < 0.05$) (Table 3). According to the original cut-off point, MNA had a sensitivity of 93.5%, specificity of 44.6% PPV 65.4%, and NPV 86% of MNA with a total diagnostic accuracy of 70.5% (Table 3).

The area under ROC curves using the serum albumin concentration level as golden standard area showed the highest values of 0.927 (Fig. 1). The AUC (95% CI) value indicates that MNA had excellent diagnostic accuracy to diagnose malnutrition with an overall accuracy of 92.7% (88.5, 96.9). In addition, Maximum Youden's J index calculated using the ROC curve was 0.804. At this Youden's index value, the newly developed optimal cut-off value for the MNA tool was 20.5 to detect the markers of malnutrition (*i.e.*, merged at risk of malnutrition and malnutrition). Based on this cut-off value, the MNA total

**Table 2  Cronbach's alpha for MNA tool applied in the elderly population aged 60 and above years.**

| Items | Cronbach's α if item deleted |
|---|---|
| Decreased food intake | 0.596 |
| Weight loss | 0.559 |
| Mobility status | 0.607 |
| Acute stress | 0.633 |
| Depression | 0.599 |
| BMI category | 0.614 |
| Living without support from other | 0.591 |
| Number of drugs per day | 0.616 |
| Ulcer on skin | 0.620 |
| Number of meals | 0.619 |
| Consumption of protein | 0.589 |
| Fruit and/or vegetable intake | 0.565 |
| Fluid intake | 0.596 |
| Feeding status | 0.576 |
| Self-perception of nutritional status | 0.567 |
| Self-perception of health status | 0.526 |
| MUAC category | 0.582 |
| CC category | 0.565 |
| Overall Cronbach's alpha | 0.605 |

**Table 3  Measure of correlation, agreement and diagnostic test between MNA and Serum albumin concentration.**

| MNA correlation by Spearman's rho ($r_s$) | | P value |
|---|---|---|
| With serum albumin concentration | 0.75 | 0.000 |
| With BMI classification of nutritional status | 0.39 | 0.000 |
| With Self-perception of nutritional status | 0.51 | 0.000 |
| MNA agreement with serum albumin | | |
| Weighted kappa (95% CI)[a] | 0.56(0.470,0.642) | |
| Weighted kappa (95% CI)[b] | 0.39(0.269,0.514) | |
| Diagnostic accuracy | | |
| Sensitivity | 0.935 | |
| Specificity | 0.446 | |
| PPV[c] | 0.654 | |
| NPV[d] | 0.86 | |
| Total Diagnostic accuracy | 0.7045 | |

**Notes.**
[a] Malnutrition, risk of malnutrition, well-nourished.
[b] Malnutrition and risk of malnutrition, well-nourished.
[c] Positive predictive value.
[d] Negative predictive value.

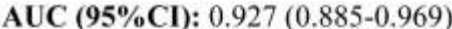

**AUC (95%CI):** 0.927 (0.885-0.969)

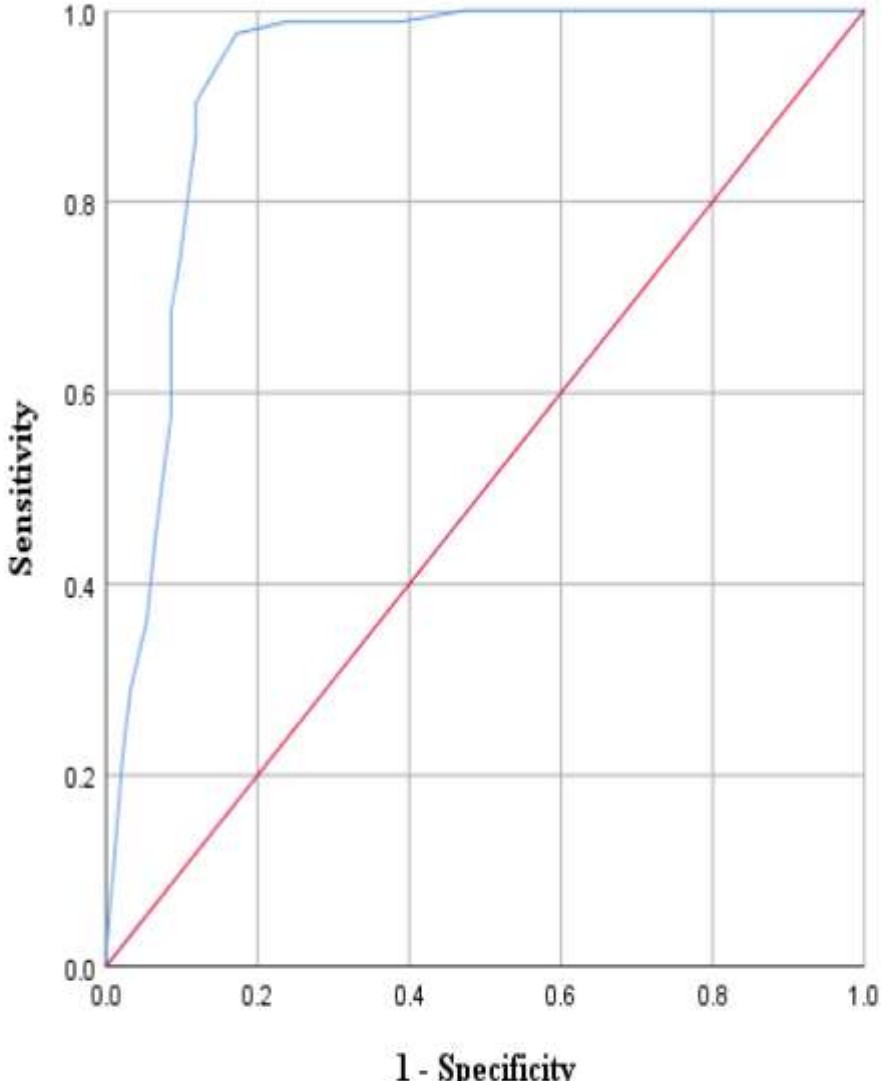

**Figure 1** **ROC curves of MNA tool.** The ROC curves of one hundred and seventy six samples for the MNA tool as compared to serum albumin concentration of participant elderly people aged 60 and above years.

score of <20.5 points, as markers of malnutrition, the sensitivity increased to 97.6%, and specificity increased to 82.8%.

## DISCUSSION

The MNA long-form had adequate predictive ability of markers of malnutrition as compared to serum albumin concentration level, among the elderly community population. It showed high sensitivity and specificity, with a somewhat modified optimal cut-off value.
Using the original cut-off value, its sensitivity, diagnostic accuracy and overall diagnostic accuracy were 93.5%, 70.45%, and 92.7% respectively. Our data also suggests that the MNA long-form is valid, with a cut-off value of 20.5, using serum albumin concentration (<3.5 g/dl) as a golden standard. This optimal cut-off value had a sensitivity of 97.6% and a specificity of 82.8%. Moreover, the MNA long-form had a strong agreement with serum albumin concentration, for identifying elderly individuals with malnutrition or those at risk of malnutrition (weighted kappa = 0.556 (0.470, 0.642). The prevalence of malnutrition using serum albumin concentration was found to be 13.1%. Compared to the gold standard, the MNA long-form overestimates malnutrition by 5.1%.

We evaluated the reliability and validity of the MNA tool among the elderly population of Ethiopia living in Meki town. Similar to other reports, we have found that the tool is easy to administer and use (*Guigoz, 1994*; *Nestlé Nutrition Institute, 2022b*; *Nestlé Nutrition Institute, 2022a*; *Guigoz, 2006*; *Guigoz & Bruno, 1995*). Our results indicated that the MNA tool had acceptable overall internal consistency with a Cronbach's alpha score of 0.605. Hence, it seems reasonable to assume that the MNA tool is reliable, measures what it intends to measure, and the scale is consistent and homogeneous, at least in our sample.

To demonstrate the validity of the MNA tool, it is common practice in the field to calculate the correlation of the MNA tool with serum albumin concentration. In this regard, a strong significant positive correlation was detected ($r_s = 0.746$), contributing to the tools' criterion-related validity. Similarly, the tool demonstrated statistically significant concurrent validity with BMI nutritional status classification and self-perceived nutritional status. Furthermore, the MNA tool had a moderate agreement with serum albumin concentration level with a weighted kappa value of 0.556. All these results indicate the appropriateness of the tool for the population under study.

Employing originally established cut-off value, the MNA had 93.5%, 44.6% and 65.4% Sensitivity, specificity and PPV respectively. The tool also demonstrated an excellent overall diagnostic accuracy with a value of 92.7%. These results are comparable with other findings conducted elsewhere. Even though studies from settings like Turkey reported that the MNA tool had a sensitivity of 92% (*Sarikaya et al., 2015*), others have reported somewhat lower values; like Spain, 85.1% (*Bleda et al., 2002*), Iran 82% (*Amirkalali et al., 2010*), Brazil 89% (*Ferreira, Nascimento & Marucci, 2008*), Japan 81% (*Kuzuya et al., 2005*), and Nepal 86% (*Ghimire et al., 2017*). If, however, we used the value reported by the MNA tool developers, the results in this study are relatively low. The originally reported values for sensitivity and specificity are 96% and 98% respectively.

This is not surprising as such variations in sensitivity and specificity could arise from the nature of the study population, sampling technique and golden standard used to validate the MNA tool.

Hence, this study supports the use of MNA as a reliable and valid tool for screening malnutrition among the elderly population living in the Ethiopian community. However, according to the newly developed best-fit cut-off value for the MNA tool (score of <20.5), the sensitivity increased to 97.6% and specificity increased to 82.8%. This new optimal cut-off value, however, decreased the diagnostic accuracy to 80.4%. Therefore, all in all, with the updated cut-off value, the sensitivity, and specificity of the MNA tool become

more comparable to the original developers (*Guigoz & Bruno, 1995*). Nevertheless, it's worth noting that the new cut-off value takes sensitivity and specificity to a much higher value than the original cut-off points. Therefore, further studies are needed to evaluate newly developed cut-off value for Ethiopian elders using combined biomarkers as the golden standard. Till then the results of this study indicate the appropriateness of the tool for the population under study.

Readers should take note of the following limitations while interpreting the results of our study. First, dietary assessment methods were not applied and indicators for micronutrient status were not assessed for the participants. Second, we only applied a single golden standard method, *i.e.*, serum albumin, to validate the tool. The correlation between the MNA tool and BMI scores should also be interpreted with caution. It is repeatedly reported that BMI is a limited application in the nutritional assessment of obese individuals but otherwise are malnourished. This may have influenced the result to some degree. Third, the sample size is low, and larger sample size could have provided more statistically robust results. Among the strengths of this study was the inclusion of the community-dwelling elderly population, and the use of a random sampling method to recruit households.

## CONCLUSION

This study indicated that the MNA tool was a valid and reliable tool for the Ethiopian elderly population in the community setting. Future studies should explore the cost-effectiveness of MNA long-form to establish if the tool is suitable for low resource settings, cost-wise.

**List of abbreviation**

| | |
|---|---|
| **AUC** | Area Under Curve |
| **BMI** | Body Mass Index |
| **CC** | Calf Circumference |
| **CI** | Confidence Interval |
| **MNA-LF** | Mini Nutritional Assessment Long-Form |
| **MUAC** | Mid-Upper Arm Circumference |
| **NPV** | Negative Predictive Value |
| **PPV** | Positive Predictive Value |
| **ROC-Curve** | Receiver-Operating Characteristic Curve |

## ACKNOWLEDGEMENTS

We are also appreciative to the participants in the study, as well as the data collectors and supervisors.

### Funding
The author received no funding for this work.

## Competing Interests

The author declares that they have no competing interests.

## Author Contributions

- Megersso Urgessa conceived and designed the experiments, performed the experiments, analyzed the data, prepared figures and/or tables, authored or reviewed drafts of the article, and approved the final draft.

## Ethics

The following information was supplied relating to ethical approvals (*i.e.*, approving body and any reference numbers):

The Jimma University, Institute of Health, Ethical review committee (ERC) approved this study. The approval number of the ERC was ERB-00063/2020.

## Data Availability

The raw data are available in the Supplementary File.

## Supplemental Information

Supplemental information for this article can be found online at http://dx.doi.org/10.7717/peerj.14396#supplemental-information.

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
