# Peer review of "The Mini Nutritional Assessment tool’s applicability for the elderly in Ethiopia: validation study"

_PeerJ, doi:10.7717/peerj.14396_

## Round 0.1 · original submission · Minor Revisions

Dear authors,
Please address the reviewers corrections below:
Reviewer 1 suggestions
Basic reporting
• The author comments on plural in several parts of the manuscript, but it is the only author present in the study. Does this manuscript contain more eligible co-authors? If yes, please include them with each contribution to the development of the manuscript and investigation.
• I would strongly suggest exchanging the keyword “MNA” for “Malnutrition” since it is the correct MeSH Term which could increase the article's visibility and searchability. Also, another keyword such as “Epidemiology” can be suitable for data libraries' increased scan.
• Page 7, lines 64-66: I would suggest using more suitable research for citing such important statistics for this manuscript. The referred citation from Morley J. is dated (1997), please update the data from the previous 5 years to increase readability.
• Overall, the context and background are sound and present consistent information for the reader, however, I believe that two main issues need to be addressed to increase the quality of the manuscript: 1) prefer using journal articles instead of books for citations and 2) update the major date for citations in all manuscript for the last 5 years. It is not a problem to use older citations, especially those that represent the state-of-the-art, but the majority of the citations are considerably dated.
Experimental design
• Page 10, line 134: please, correct a typo of “cupper-and-zinc-free syringe” for “copper and zinc-free syringe”.
• Please provide the mean intra-evaluator error for body composition measurements (perimeters, weight and height). The error can be calculated post-evaluation and can increase the robustness of the method since the statistical section is sound.
• In the methods section, there is no information on how the self-perception nutritional status was measured/determined. Since it is an important parameter for the validation of MNA in the Ethiopian elderly population, I recommend including this information in the appropriate section of the manuscript.
Validity of the findings
• The study is well designed and includes appropriate statistical analyses for its main hypothesis. The points that need more information to achieve maximum replicability and detail I have already mentioned in my review

Reviewer 2 suggestions
Line 113: ‘In Ethiopia, MNA has not been tested on the elderly population’ are you sure with this statement? In Ethiopia many studies were conducted to validate this tool. So, I suggest that rather than saying this try to write the new thing you addressed with current study; like using serum albumin as gold standard which is the strength in your case.
 Try to explain in detail the way you calculated sample size to make it clear.

·

Basic reporting

• The author comments on plural in several parts of the manuscript, but it is the only author present in the study. Does this manuscript contain more eligible co-authors? If yes, please include them with each contribution to the development of the manuscript and investigation.
• I would strongly suggest exchanging the keyword “MNA” for “Malnutrition” since it is the correct MeSH Term which could increase the article's visibility and searchability. Also, another keyword such as “Epidemiology” can be suitable for data libraries' increased scan.
• Page 7, lines 64-66: I would suggest using more suitable research for citing such important statistics for this manuscript. The referred citation from Morley J. is dated (1997), please update the data from the previous 5 years to increase readability.
• Overall, the context and background are sound and present consistent information for the reader, however, I believe that two main issues need to be addressed to increase the quality of the manuscript: 1) prefer using journal articles instead of books for citations and 2) update the major date for citations in all manuscript for the last 5 years. It is not a problem to use older citations, especially those that represent the state-of-the-art, but the majority of the citations are considerably dated.

Experimental design

• Page 10, line 134: please, correct a typo of “cupper-and-zinc-free syringe” for “copper and zinc-free syringe”.
• Please provide the mean intra-evaluator error for body composition measurements (perimeters, weight and height). The error can be calculated post-evaluation and can increase the robustness of the method since the statistical section is sound.
• In the methods section, there is no information on how the self-perception nutritional status was measured/determined. Since it is an important parameter for the validation of MNA in the Ethiopian elderly population, I recommend including this information in the appropriate section of the manuscript.

Validity of the findings

• The study is well designed and includes appropriate statistical analyses for its main hypothesis. The points that need more information to achieve maximum replicability and detail I have already mentioned in my review

·

Basic reporting

The document tried to use clear, and unambiguous English. The structure conforms to the journal standards. Figures are relevant, high quality and well labelled & described.

Experimental design

The title is in line with the scope of the journal. the research question is well defined relevant and meaningful.
Vigorous investigation performed to a high technical & ethical standard.
Methods tried to be described with sufficient detail & information to replicate.
Just to make it sound I want to give the following comments and suggestions here.
Line 113: ‘In Ethiopia, MNA has not been tested on the elderly population’ are you sure with this statement? In Ethiopia many studies were conducted to validate this tool. So, I suggest that rather than saying this try to write the new thing you addressed with current study; like using serum albumin as gold standard which is the strength in your case.
 Try to explain in detail the way you calculated sample size to make it clear.

Validity of the findings

No comment

Additional comments

The study tried to validate mini nutritional assessment tool among 176 elders in Meki town. Reliability of the tool was checked. Sensitivity, specificity, positive predictive value (PPV), and negative predictive value (NPV) were calculated using serum albumin concentration as a golden standard. In fact it is an interesting study which contributed to fulfill the knowledge gap in this area in Ethiopian context. Further, the results can give interesting comparative data, thus allowing a better knowledge of nutritional status variations in the elderly.

---

## Round 0.2 · accepted · Accept

The revised version of the manuscript has improved significantly. More information was added and the author answered all reviewers' comments.

·

Basic reporting

The revised version of the manuscript has improved significantly. More information was added and my comments were all answered by the author.

Experimental design

Adequate design to test the hypothesis. There are no pending issues in my view.

Validity of the findings

There are no pending issues in my view.